# Does Context Matter? Effective Deep Learning Approaches to Curb Fake News Dissemination on Social Media

Jawaher Alghamdi [1,2,*], Yuqing Lin [1] 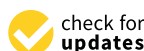 and Suhuai Luo [1]

1   School of Information and Physical Sciences, College of Engineering Science and Environment,
    The University of Newcastle, Callaghan, Newcastle, NSW 2308, Australia
2   Department of Computer Science, King Khalid University, Abha 62521, Saudi Arabia
*   Correspondence: jawaher.alghamdi@uon.edu.au

**Abstract:** The prevalence of fake news on social media has led to major sociopolitical issues. Thus, the need for automated fake news detection is more important than ever. In this work, we investigated the interplay between news content and users' posting behavior clues in detecting fake news by using state-of-the-art deep learning approaches, such as the convolutional neural network (CNN), which involves a series of filters of different sizes and shapes (combining the original sentence matrix to create further low-dimensional matrices), and the bidirectional gated recurrent unit (BiGRU), which is a type of bidirectional recurrent neural network with only the input and forget gates, coupled with a self-attention mechanism. The proposed architectures introduced a novel approach to learning rich, semantical, and contextual representations of a given news text using natural language understanding of transfer learning coupled with context-based features. Experiments were conducted on the FakeNewsNet dataset. The experimental results show that incorporating information about users' posting behaviors (when available) improves the performance compared to models that rely solely on textual news data.

**Keywords:** fake news; misinformation; deep learning; BERT

## 1. Introduction

Online social networks (OSNs) have gained importance due to their easy accessibility. They are tools used for exchanging information and influencing public opinion, rather than just a means of communicating between individuals [1]. In recent years, fake news has spread more widely due to the ease with which it can be created and distributed online. This type of news is not actual news, it is fake news that is made real for a specific purpose [2].

Twitter is a common social media platform that people use to express their opinions and share their ideas with others. User-generated content (UGC) on OSNs, especially Twitter, is gaining more importance in the research community owing to its value in discovering patterns that significantly benefit various applications. According to the fake news velocity study, tweets that include falsified information reach people six times faster than tweets that include trustworthy information [3]. This indicates how terribly fake news disseminates and how it can have adverse social effects. The concern lies in the quick reactions, such as retweets, likes, and shares of a tweet (fake news story) received on Twitter without pre-thinking, aggravating the problem even more. According to [4], false news (particularly political news) on Twitter is usually retweeted by more users and spreads extremely rapidly. As Wang et al. [5] pointed out, the proliferation of fake news on social media begins with user-posting behaviors.

Much of the current work on fake news detection focuses on textual news content (see Section 2 for more details), ignoring user behavior clues that could have the potential to advance fake news detection performance. Moreover, most of these studies are based on extensions of the bag-of-words representation or other context-independent embeddings,

which cannot capture deep semantically contextualized information. The ability to capture and understand semantics and contextualized information about the input text is crucial to identifying fake news.

Given the benefits of user behavior clues in identifying fake news, we argue that the interplay between social user engagements (via feature engineering) and feature learning derived from state-of-the-art (SOTA) deep context-aware representation methods (such as BERT) has the potential to boost fake news classification performance. Given the power of user-behavior cues and the characteristics of SOTA pre-trained language models (PLMs) in generating deep-semantical contextual patterns of a given input text (i.e., news article), their complementary effects need to be investigated.

As such, in this study, we investigated the impacts of user posting behaviors on fake news detection using the FakeNewsNet dataset and explored two research questions: (1) Does modeling both news content and user-posted behavioral cues improve fake news detection performance? and (2) How effective are user-posted behavioral cues in improving the detection performance of the proposed deep learning frameworks?

The key contributions of this paper are as follows:

- An examination of the performance of several deep learning algorithms and the current state-of-the-art word embeddings, such as BERT, on a benchmark dataset of fake news.
- For detecting fake news, new hybrid CNN-RNN architectures using attention modules were developed.
- Extensive experiments on two real-world fake news datasets demonstrate the effectiveness of the proposed frameworks for detecting fake news.

The rest of this paper is organized as follows. Section 2 summarises the literature related to the detection of fake news. Section 3 provides an overall background of the models we used. We describe the problem statement in Section 4. The proposed deep-learning approaches are described in Section 5. Experiments on the performance of the predictive models are presented in Section 6. The experimental results are discussed in Section 7. Finally, Section 8 concludes the paper.

## 2. Related Works

Current research in fake news detection can be generally classified into content- and context-based fake news detection. According to [6], the former takes into account text-based features, such as general features, which are catalysts for describing content style from four linguistic levels: lexicon, syntax, discourse, and semantics. Some statistical techniques are applied to the lexicon level, such as the bag-of-words (BoW) model [7]. At the syntax level, part of speech (POS) (e.g., nouns and verbs) frequency is assessed using POS taggers [7]. A rhetorical structure theory and rhetorical parsing tool can be used at the discourse level to capture the resulting frequency of rhetorical relations among sentences as features [7]. On the semantic level, these frequencies correspond to lexicons or expressions assigned to each psycholinguistic category [8] (e.g., LIWC [9]). A second type of text-based feature is called a latent textual feature, which is used for news text embedding. These features can be conducted either at the word level [10], sentence-level [11], or document-level [11], which result in dense vector representations that can then be processed further. Some of the studies on detecting fake news are mainly based on the content; in [12], to detect fake news, the authors applied CNN and BiLSTM to embed textual and speaker metadata. In [13], the authors presented twenty-six linguistic-based textual features to detect fake news. In [9], the authors employed linguistic features (e.g., punctuation, readability, syntax, and psycholinguistic features) to distinguish between true and fraudulent news items, while the authors of [14] developed an enhanced set of linguistic features for distinguishing fake from true news.

In [15], a Chinese WeChat clickbait dataset was created, and a multi-featured method of detecting WeChat clickbait was proposed, using semantic, syntactic, and auxiliary information. MFWCD-BERT and MFWCD-BiLSTM are models based on the MFWCD

framework with varying parameter scales, which use Bidirectional Encoder Representation from Transformers (BERT) and lightweight Bidirectional Long Short-Term Memory (Bi-LSTM) networks with attention mechanisms to encode headline semantics, respectively. The researchers also presented an improved Graph Attention Network (GAT) with attention mechanisms to capture valuable structures in titles based on local syntactic structures. MFWCD performs better than compared baseline methods in clickbait detection, proving its effectiveness and interpretability. For other related research conducted for clickbait detection, see [16,17].

It is inherently difficult to detect fake news because fake news is usually written intentionally to mislead the reader. Among the potential clues to the detection is the context-based features (aka auxiliary contextual information), such as user interactions on online social networks. Researchers combined content- and context-based features for characterizing fake news. For example, Mouratidis et al. [18] combined network account features with linguistic features. Shu et al. [19] developed the social article fusion (SAF) model, which combines social engagement features with linguistic aspects using the FakeNewsNet dataset. Authors in [20] presented a deep learning model based on the hierarchical attention network using news content and user comments on Twitter for detecting fake news, and their framework achieved state-of-the-art results in the FakeNewsNet dataset.

In the context of Twitter, post-based features refer to those retrieved from source tweets. Several studies used contextual information from tweets, such as temporal patterns embedded in a series of replies (i.e., comments) posted by users on social media and other elements representing their interactions and engagements. Several studies have used similar temporal patterns to identify useful patterns for fake news detection. For example, Ma et al. [21] proposed a technique called SVM-TS that employs time series of aggregated news attributes to detect fake news. Linguistic-based features can also be extracted from each tweet. Furthermore, features extracted from the topic using topic modeling approaches, such as Latent Dirichlet Allocation (LDA) [21]. Credibility features can also be extracted for posts in order to assess the reliability degree [22]. Using user-based clues could help differentiate between fake and real news. This is due to the fact that individuals who are more likely to spread incorrect information have distinct characteristics and attributes than those who are not [23]. These findings prompted researchers to look into user-based features for detecting fake news. For example, in [23], the authors examined user profiles and extracted features to distinguish between fake and real news. Wang [12] uses user profile features as input to a hybrid CNN model for fake news detection. In [24], the authors concluded that the idea of incorporating a speaker profile could noticeably enhance performance. Most importantly, social propagation features, such as followers and retweet counts, can be used as useful insights for determining how fake news can spread in social media [19].

In recent research, many useful methods for detecting fake news have relied on sequential neural networks for encoding news content and social context information, in which text sequences were analyzed unidirectionally [25]. With this bidirectional training approach, long-distance and semantic dependencies can be captured in the given sentences.

A deep learning approach (FakeBERT) is proposed in [25] that uses BERT (Bidirectional Encoder Representations from Transformers) and the parallel blocks of a deep Convolutional Neural Network (CNN) with varying kernel sizes and filters combined with BERT. They found that this combination is useful in dealing with ambiguity, which is the most difficult challenge in natural language understanding. However, the authors overlooked the benefit of modeling user behavior clues, which could improve classification performance. Using the LIAR dataset, a machine learning framework for automatic fake news detection using $BERT_{base}$ was presented by Alghamdi et al. [26]. In order to extract local features, the input text was encoded using $BERT_{base}$, and then the output sequence was fed into a CNN network. Then, a CNN followed by a BiLSTM was used to encode the user profile features. The final output of the two components is fed into a classification layer. The authors concluded that user behavior features are important for fake news detection.

A comparative study using different machine learning and deep learning models was conducted [27]; the authors showed the power of BERT and its variations in improving detection performance across different datasets.

In a study conducted by Natali Ruchansky et al. [28], the researchers analyzed social media data and developed a hybrid deep learning model that showed an accuracy of 0.892 on Twitter data and 0.953 on Weibo data. The researchers concluded that capturing the temporal behavior of articles and learning source characteristics about users' behavior was important for detecting fake news. Exploiting useful clues from user behavior on social media is of great importance to detect fake news, and as such, in this work, we investigate the impact of user posting behavior for fake news detection using the FakeNewsNet dataset. In addition, we assess the effectiveness of a different set of user posting behavior features. A majority of existing studies used features extracted from a given news article to model fake news detection. This approach is reasonable since news text is most commonly provided in isolation in public datasets; however, content-based features are often insufficient since malicious entities are frequently manipulating content, following success at detection, to mimic trustworthy content. A natural question arises: Can associated user posting behavior information on social media platforms (e.g., Twitter) improve the performance of fake news detection?

To this end, noting the power of user behavioral features, this study focused on leveraging joint learning of news content and the user posting behavioral clues on Twitter for fake news detection. We hypothesize that news content and the user posting behavior attributes contain complementary information that needs to be encoded and captured simultaneously for fake news detection. Furthermore, the majority of existing studies modeled text input using representations extracted from pre-trained context-free embedding models, such as GloVe, to train fake news detection models. A key limitation of this process is that it assigns fixed vector values to each word in a given sequence regardless of its context. Progress in the realm of natural language processing has centered on deep contextualized models that leverage the deeper long-term semantical contextual relationships between words and sentences in a given corpus. In this work, we leveraged the power of context-aware embedding models, such as BERT, to encode the news text. Generally speaking, the textual news content is initially encoded using BERT to extract deep contextual representations, followed by a deep learning architecture. Next, user features are processed by stacking BiGRU layers. These two components are concatenated to combine the features extracted from news articles and their associated user behavior clues into a single feature vector. This combined feature vector is then fed into a classification layer with a single neuron, activated using a sigmoid function.

## 3. Background: Models

### 3.1. Word Embedding

An embedding is typically learned from a large corpus of text. In NLP, there are two main alternatives for obtaining a distributed representational vector that captures the semantics of the input text.

- A word-based representation.
- A context-based representation.

### 3.1.1. Word-Based Representations (Non-Contextualized Embeddings)

Context-independent methods produce the same embedding vector for each word in the vocabulary, irrespective of its context. Examples of the most common context-independent pre-trained embeddings include:

- A 300-dimensional vector was generated by **Word2Vec.** using a large corpus of news articles with 300 million tokens.
- Pre-trained **GloVe** used 27 billion tokens in a huge corpus of tweets, resulting in a 200-dimensional vector.

3.1.2. Context-Based Representations (Contextualized Embeddings)

By capturing only short-range co-occurrence context, non-contextual embedding models are unable to capture deeper contextual relationships. In contrast, context-aware embedding models can compute the embedding for a word based on its context. An example of the most common deep contextualized pre-trained embedding models is BERT. In Devlin et al. [29], the bidirectional encoder representation from transformer (BERT) is introduced as the first deeply bidirectional, unsupervised language representation, which functions as a transformer encoder with multiple layers (performs self-attention in both directions) that conditions both left and right contexts simultaneously. Therefore, BERT generates embeddings that are context-aware.

Furthermore, to remove the unidirectionality constraint, BERT performs pre-training using an unsupervised prediction task, which includes a masked language model (MLM) in charge of understanding context and then making predictions (of words). Thus, the model can produce a vector representation that can capture the general information of the input text. These semantic representations of each word in the input text can be improved using an attention mechanism when different words in a context show different effects in boosting semantical representation. As a core component of transformer architecture, the attention mechanism's underlying role is to assign less or more weights to different parts of text towards the output (i.e., differentiate the contribution of different parts of the input on the output). Attention can be considered as a function that maps queries and follows key-value and output vector pairs; the scaled dot-product attention formula can be seen in Equation (1).

$$Attention(Q, K, V) = Softmax(\frac{QK^T}{\sqrt{d_k}})V \tag{1}$$

Query, key, and value are marked by the letters $Q$, $K$, and $V$, respectively. $\sqrt{d_k}$ represents the dimension of the key vector $k$ and query vector $q$. A Softmax activation function is used by attention to normalize the inputs to a value between 0 and 1. Although BERT uses a transformer's encoder, it employs a multi-head attention mechanism, as shown in Equation (2), where each distinct head and its corresponding weight matrices are identified by the subscript $i$.

$$MultiHead(Q, K, V) = Concat(head_1, \ldots, head_h)W^O \tag{2}$$

where each $head_i$ is calculated using the formula below, and $W^O$ represents a weight matrix that was trained along with the model:

$$head_i = Attention(QW_i^Q, KW_i^K, VW_i^V) \tag{3}$$

Even though BERT contains millions of parameters (i.e., BERT$_{base}$ contains 110 million parameters and BERT$_{large}$ has 340 million parameters) [29], it is relatively inexpensive to apply BERT to downstream tasks if the parameters are fine-tuned using a pre-trained model. In this work, we use BERT$_{base}$.

*3.2. CNN*

Convolutional neural networks (CNNs) have demonstrated effectiveness in several NLP tasks in which one-dimensional convolutional neural networks (Conv1D) are used. As a result, Conv1D iterates over training data using a filter with a fixed window size (each filter cell is initially set to a weight) and uses the sliding window to iterate over the training data. At each step, the inputs (word vectors) are multiplied by these filter weights to generate a feature map (filter output array) that encodes informative features derived from input training data. A CNN is well known for its ability to automatically extract local features and capture them more accurately. As such, CNNs are generally better at extracting features from input data, thereby reducing dimensionality and increasing robustness [30].

*3.3. BiGRU*

A variant of the gated recurrent unit (GRU) consists of only two gates: an update gate that acts as a catalyst for combining the forget and input gates and controls the information flowing to the current state, and a reset gate that determines when to ignore the previous hidden state [31]. The update and reset gates are computed similarly to LSTM as follows [31]:

$$r_t = \delta(W_r h_{t-1} + U_r x_t + b_r), \tag{4}$$

$$z_t = \delta(W_z h_{t-1} + U_z x_t + b_z), \tag{5}$$

$$h_t = (1 - z_t) \odot h_{t-1} + z_t \odot \tilde{h}_t, \tag{6}$$

$$\tilde{h}_t = tanh(W_{\tilde{h}_t}(h_{t-1} \odot r_t) + U_{\tilde{h}_t} x_t). \tag{7}$$

In the formulas above, $\delta(.)$ signifies the logistic sigmoid function, $W$ and $U$ are gate weight matrices, and $h_t$ and $b$ are the hidden state and bias vectors, respectively. Basic RNNs simply consider the previous context and lack the ability to capture future context. As a result of the breakthrough design, bi-directional long short term memory (BiLSTM) and bi-directional gated recurrent unit (BiGRU) are suitable options for accounting for the future and preceding contexts. To do this, the forward and backward hidden layers are integrated, controlling the temporal information flow in both directions and leading to better learning.

*3.4. Attention Mechanism*

Recently, the attention paradigm, inspired by human biological systems, has gained traction in natural language processing (NLP) fields. The NLP community is preparing for the paradigm shift by designing models that can assign weights to various parts of a given input text, capturing more relevant information for further processing. The attention model aims to mimic humans' biological systems, where, given a piece of text, humans can selectively identify what is most vital and relevant in a given context while ignoring irrelevant information. As a consequence, certain parts of the input can be considered adaptively by the model [32]. Furthermore, the attention mechanism enables the model to learn to attend to the relevant parts of the given input (e.g., hidden states of a BiGRU network) to generate a single salient vector representation. In light of this, the model may identify the words that are being concentrated on for each input text by "attending to" particular areas of the input when processing the data. Typically, weighting different words in a sentence are done by combining all hidden states, generating a single vector representation, as described below.

$$\alpha_t = \frac{\exp(v^T \cdot \tilde{h}_t)}{\sum_t \exp(v \cdot \tilde{h}_t)} \tag{8}$$

$$S_{Aw} = \sum_t \alpha_t h_t \tag{9}$$

where $v$ is a trainable parameter [33] and the hidden states are computed in Equations (6) and (7).

**4. Problem Definition**

We approach the challenge of fake news detection as a binary classification, in which a model is built to predict whether a piece of news is fake or real based on the provided attributes. Formally, assume $A = \{a_1, a_2, \ldots, a_l\}$ is a set of news articles (where $l$ is the length of the input) and $U = \{u_1, u_2, \ldots, u_n\}$ represents user posting behavior informa-

tion towards such news (e.g., the number of retweets such a news article received on Twitter). Given news articles and the associated user-posted behavioral cues, the purpose of fake news identification is to determine whether the input text represents fake or legitimate content.

That is, $f : f(A, U) \rightarrow$ {fake, real} $\ni$,

$$
F(A) = \begin{cases} fake, & \text{if } A \text{ is fake content,} \\ real, & \text{otherwise.} \end{cases} \tag{10}
$$

## 5. The Proposed Approaches

### 5.1. BERT$_{base}$-CNN-BiGRU-ATT

In this subsection, we will introduce the first deep learning architecture proposed in this paper for fake news detection. Specifically, as illustrated in Figure 1, we first perform BERT$_{base}$ on input text—suppose $x_{ij} \in \mathbb{R}$, represents the d-dimensional word embedding of the $j$-th word in the $i$-th sentence, to obtain deep contextualized representations since BERT$_{base}$ plays a bidirectional attention mechanism on input text, ensuring better coverage of global semantic understanding. Then, a CNN layer is applied to capture local features as follows. Given a sentence, assume $X_{1:i:m} = x_{11} \oplus x_{12} \oplus \ldots x_{1m}$, where $m$ denotes the length of a given sentence $i$ and $\oplus$ refers to a concatenation operation. A feature map $F$ is generated by performing a convolutional operation with a filter $T \in \mathbb{R}^{h \times d}$ on $X_{i:i+h-1}$ using the equation $f_i = \sigma(T \cdot X_{i:i+h-1} + b)$ where $\sigma$ represents a non-linear activation function and $b$ denotes a bias term. The final feature map $F$ is obtained by applying the previous step to all sliding windows of $h$ words in a sentence. This $F$ is fed into a max-pooling layer and to a global average pooling layer for capturing the most relevant and salient information, generating $F_m = [F_{m1}, F_{m2}, \ldots, F_{mk}]$ and $F_g = [F_{g1}, F_{g2}, \ldots, F_{gk}]$, respectively, with $k$ represents the number of filters used. The resulting feature map is max-pooled and global averaged-pooled. In addition, BiGRU is applied to the output of the CNN layer. The BiGRU layer consists of a forward $\overrightarrow{GRU}$ and a backward $\overleftarrow{GRU}$ to process the information from both left-to-right and right-to-left directions.

$$
\overrightarrow{h_{ij}} = \overrightarrow{GRU}(F, h_{i(j-1)}), j \in \{1, \ldots, m\} \tag{11}
$$

$$
\overleftarrow{h_{ij}} = \overleftarrow{GRU}(F, h_{i(j-1)}), j \in \{m, \ldots, 1\} \tag{12}
$$

We denote the combination of the hidden state obtained from both the forward and backward GRUs as $h_{ij} = [\overrightarrow{h}_{ij} \oplus \overleftarrow{h}_{ij}]$, which, in turn, captures full salient information about a given input text. In order to better capture salient and relevant features from the final output of BiGRU networks, the attention mechanism (see Equations (13)–(15)) is applied to the resultant output of BiGRU and the models, to assign different weights to different parts of the given input.

$$
u_{ij} = tanh(W_w h_{ij} + b_w) \tag{13}
$$

$$
\alpha_{ij} = \frac{\exp(u_{ij} u_w)}{\sum_j \exp(u_{ij} u_w)} \tag{14}
$$

where $u_{ij}$ is a hidden representation of $h_{ij}$, $u_w$ is a trainable parameter that is randomly initialized during the course of training, and the result of the product $(u_{ij} u_w)$ is used to calculate the importance score of word $j$, which is then normalized using the softmax function, generating the importance weight $\alpha_{ij}$ for each word.

Finally, the weighted sum of the concatenated word representations with the hidden states $h_{ij}$ form the news vector $n_i$.

$$
n_i = \sum_t \alpha_{ij} h_{ij} \tag{15}
$$

Ultimately, we concatenate the final output as follows: $C = F_m \oplus F_g \oplus n_i$, where $\oplus$ denotes the concatenation operation. We encode user posting behavior features using BiGRU. Assume $U$ is the final representation of users encoded features, a concatenation layer is used to extract the final informative features of the two components (news text and user posting behavior features) as follows: $O = U \oplus C$, where $O$ is the final output representation of the given news article with its associated user-posted behavioral cues. Finally, the output is passed to a classification layer with one unit, activated using the sigmoid function.

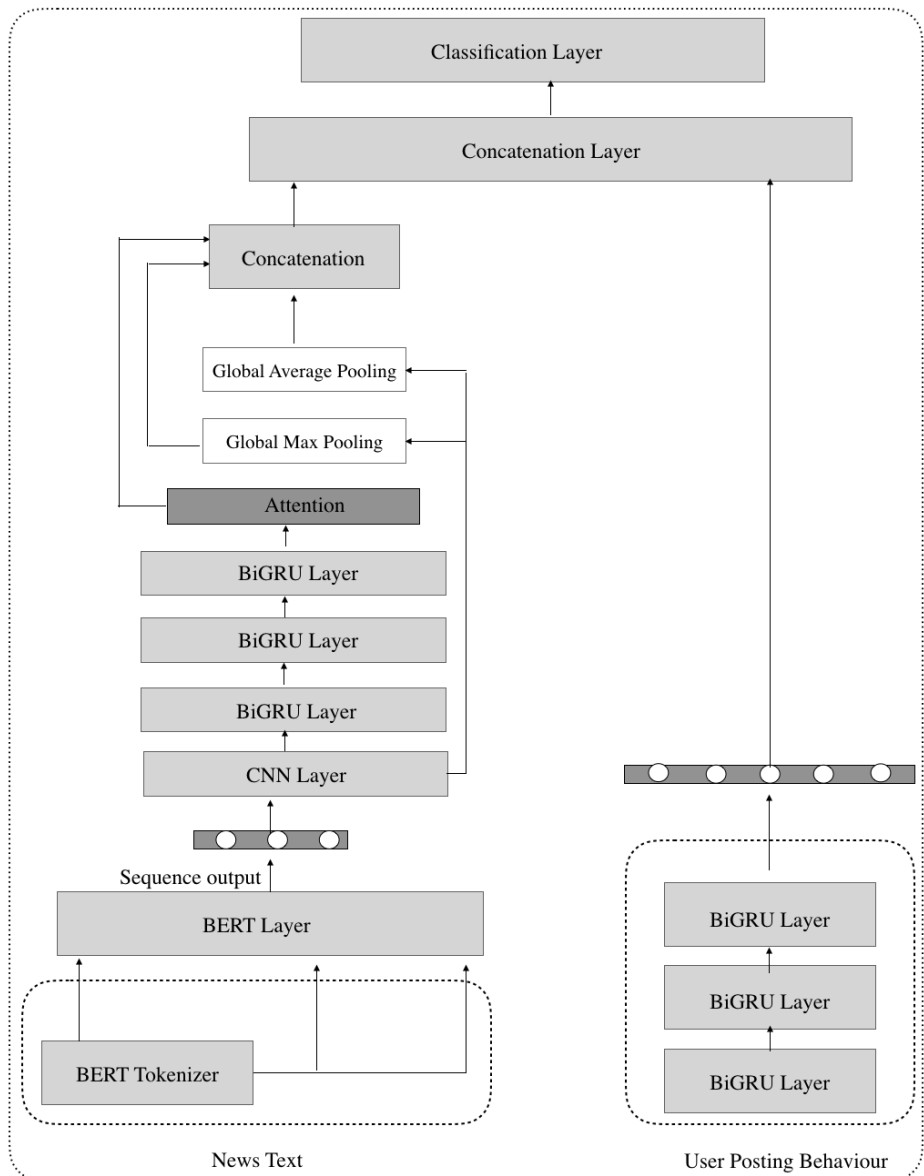

**Figure 1.** BERT$_{base}$-CNN-BiGRU-ATT architecture.

### 5.2. BERT$_{base}$-BiGRU-CNN-ATT

Here, we describe the second proposed model, which we call BERT$_{base}$-BiGRU-CNN-ATT. The architecture can be seen in Figure 2. Given a sentence $S = \{w_1, w_2, \ldots, w_N\}$, similar to the CNN-BiGRU-ATT model, BERT$_{base}$ is used to assign each word $w_i$ deep contextualized representation $X = \{x_1, x_2, \ldots, x_N\}$, which is then processed by a BiGRU, which consists of a forward $\overrightarrow{GRU}$ and a backward $\overleftarrow{GRU}$, where combining the resultant annotations forms the final representation $h_t$. The CNN architecture was used to encode

the final representation of BiGRU. The CNN architecture we used here was adapted from the research of [34].

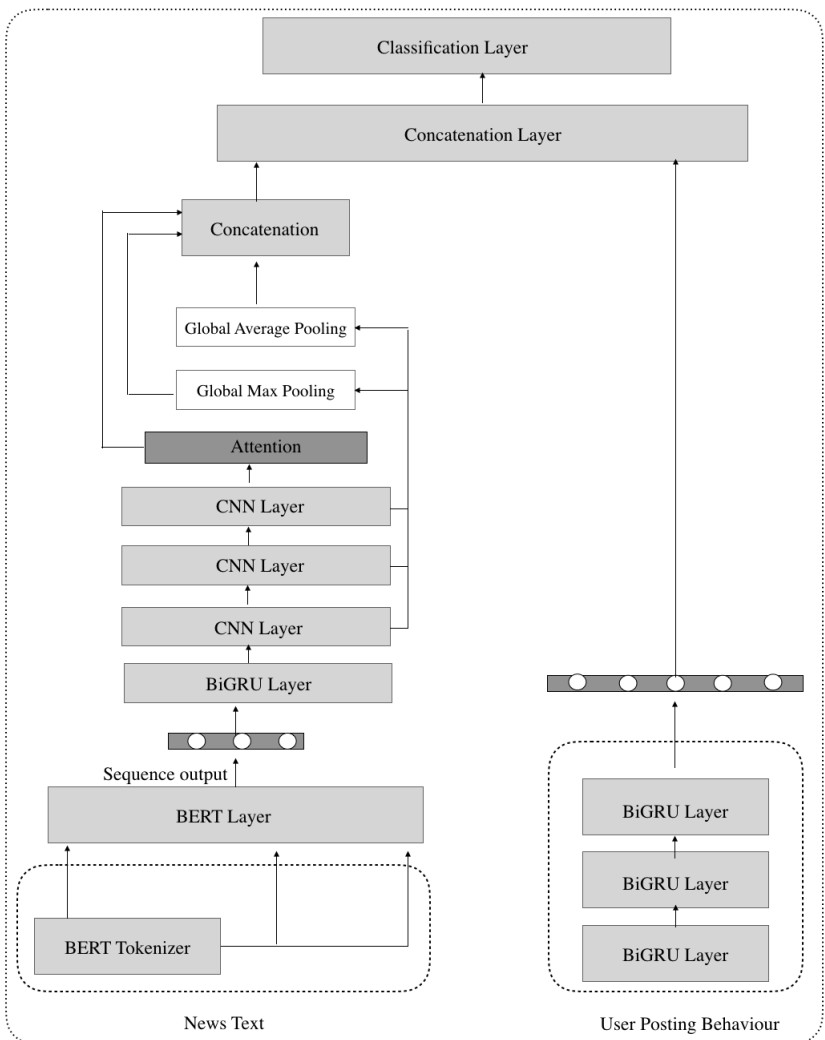

**Figure 2.** BERT$_{base}$-BiGRU-CNN-ATT architecture.

Each convolutional layer is connected to max-pooling and global average pooling layers for capturing rich–local semantic information. Following the multi-layer CNN is an attention layer that helps reflect the correlation between features by assigning different importance scores to different parts of a given input. After obtaining the output of the attention layer, a concatenation layer is used to concatenate the outputs of the max-pooling and global average pooling layers with the attention layer's output. Next, the user posting behavior features are encoded using BiGRU layers stacked on top of each other and the resulting output is concatenated with the output of the above-mentioned concatenation layer; the final output is passed to a classification layer with a single unit activated with a sigmoid function.

*5.3. BERT$_{base}$-CNN-BiGRU*

Recently, the authors in [35] showed the effectiveness of joint learning of different deep learning models, including BERT, CNN, and BiGRU networks. Inspired by this work, we investigate the effectiveness of exploiting the corresponding representation generated by the (CLS) special token in the BERT$_{base}$ model, where this token returns a vector that carries the meaning of a full sentence. The resulting representations of BERT$_{base}$ are used as input to a CNN layer, followed by max-pooling and global average pooling layers in the first

module. A BiGRU layer is also performed on the output of BERT$_{base}$, and a concatenation layer is used to concatenate the outputs of both the max- and global average-pooling layers with the pooled output of BERT$_{base}$ and the resulting representation of BiGRU. In the second module, user posting behavioral features are encoded in the same way as described with the previous model; the final output is concatenated with the output of the first module and the output is passed into a classification layer. The architecture can be seen in Figure 3.

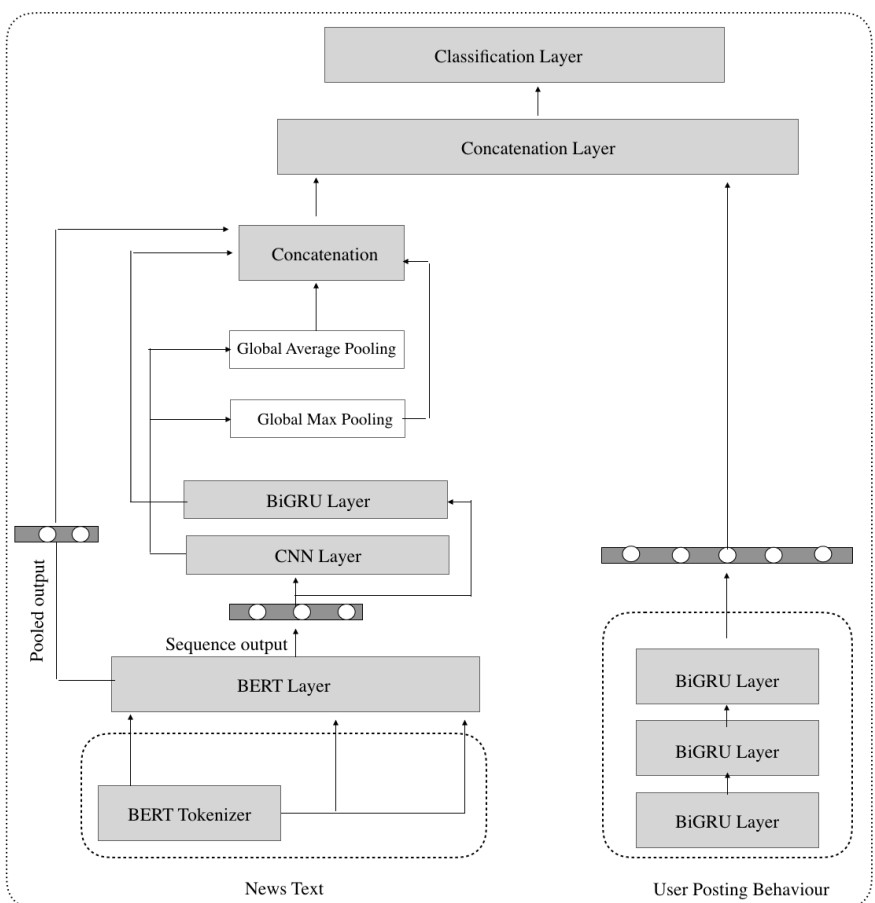

**Figure 3.** BERT$_{base}$-CNN-BiLSTM architecture.

## 6. Methodology

### 6.1. Experimental Setup

This subsection presents the experiments to assess the effectiveness of the proposed architectures. Python was used to implement the experiments, and the TensorFlow library was used to build the models. We used the pre-trained BERT$_{base}$ available on TensorFlowHub (https://www.tensorflow.org/hub, accessed on 2 January 2023). We used a macOS computer with an Intel Core i5 processor running at 2.3 GHz and 8 GB of memory. The source code is available (https://github.com/Jal-ghamdi/Context_Aware_Fake_News_Detection, accessed on 2 January 2023.)

### 6.2. Evaluation Metrics

In order to evaluate the performance of the models, we used the following four evaluation criteria: accuracy, precision, recall, and F1 score (calculated as in the equations below):

- *Accuracy* (A): is a measure of a classifier's ability to correctly identify information as false or true. Equation (16) can be used to calculate the accuracy:

$$Accuracy = \frac{TP + TN}{TP + TN + FP + FN} \tag{16}$$

- *Precision* (P): is a metric that measures the exactness of the classifier, with a low value indicating a high amount of false positives. The precision is determined using Equation (17) and indicates the number of positive predictions divided by the total number of positive class values predicted.

$$Precision = \frac{TP}{TP + FP} \tag{17}$$

- *Recall* (R): is calculated by dividing the total number of true positives by the total number of true positives and false negatives, as shown in Equation (18); it is thought to be a measure of a classifier's completeness (for example, a low recall value suggests a high number of false negatives).

$$Recall = \frac{TP}{TP + FN} \tag{18}$$

- *F*1 *score* (F1): Equation (19) is used to calculate the *F*1 *score* as the weighted harmonic mean of the classifier's precision and recall measures.

$$F1 = \frac{2 * Precision * Recall}{Precision + Recall} = \frac{2 * TP}{2 * TP + FP + FN} \tag{19}$$

where $TP$, $TN$, $FP$, and $FN$ represent true positive, true negative, false positive, and false negative, respectively.

### 6.3. Datasets

We empirically investigated our research questions using the FakeNewsNet dataset. FakeNewsNet (https://github.com/KaiDMML/FakeNewsNet, accessed on 15 January 2023) is a comprehensive dataset collected from two fact-checking platforms: PolitiFact and GossipCop. The news content is labeled (e.g., articles) and the context information is provided (e.g., user responses to the news item on social networks, such as Twitter). Table 1 shows the statistics of FakeNewsNet dataset.

**Table 1.** The statistics of FakeNewsNet dataset.

| Dataset | PolitiFact | GossipCop |
|---|---|---|
| # Candidate news | 694 | 18,676 |
| # True news | 356 | 14,129 |
| # Fake news | 338 | 4547 |

### 6.4. Compared Fake News Detection Methods

In this study, we compared our proposed methods with the state-of-the-art algorithms to assess the predictive power and viability of the proposed features for fake news detection.

- **SAF [19]:** A model that combines social engagement features with linguistic aspects using the FakeNewsNet dataset.
- **BiLSTM-BERT [36]:** The natural language inference approach (i.e., inferring the veracity of the news item) uses BiLSTM and BERT embeddings in the PolitiFact dataset.
- **LNN-KG [37]:** A model trained in the PolitiFact dataset using both textual patterns and embeddings of concepts in the input text.
- **Logistic regression (N-Gram) [38]:** N-gram-based logistic regression model for fake news detection.
- **dEFEND [20]:** A framework that comprises an encoder for news content, an encoder for user comments, and a model of co-attention for sentence–comment interaction to detect fake news.

## 7. Results

*Fake News Detection Performance*

User behavior clues are important for detecting fake news [19]. Fake news detection has previously been achieved using word embeddings and bag-of-words (see Section 2 for more details). However, neither of these representations can capture contextual information. As opposed to that, our system uses highly sophisticated representations based on bidirectional encoder representations from transformers (BERT) [29]. By training a transformer bidirectionally, BERT can learn contextual relationships among words. In addition, BERT reads the entire word sequence at once, in contrast to earlier systems that either looked at a text sequence from left to right or combined left-to-right and right-to-left training. This trait enables the model to understand a word's context based on all of its surroundings. The tables below present the results of the proposed models and the official baseline models in the two datasets. The best performance scores are highlighted in bold.

Modeling user posting behavior information in addition to news text was effective for detecting fake news. The test results in the PolitiFact dataset shown in Table 2 demonstrate that the highest-performing model is BERT-CNN-BiGRU-ATT, which outperformed the other official baselines. The $BERT_{base}$-BiGRU-CNN-ATT model obtained 87.05% accuracy, 86.90% for precision, 91.25% for recall, and 89.02% for the F1 score. The $BERT_{base}$-CNN-BiGRU model obtained a performance of 93.59% for precision, 91.37% for accuracy, 91.25% for recall, and 92.41% for the F1 score. Summaries of the prediction results of each model in the PolitiFact dataset are shown in the forms of confusion matrices in Figures 4–6. Regarding PolitiFact, all models showed similar confusion matrix results, with $BERT_{base}$-CNN-BiGRU-ATT yielding the best results. Figure 7 shows the overall scores across all of the models. The comparison of results of the GossipCop dataset (using the combination of user-behavior cues with news text) is described in Table 3 and shows that the $BERT_{base}$-BiGRU-CNN-ATT model outperformed the existing baselines and other counterpart models. The test results show that the $BERT_{base}$-BiGRU-CNN-ATT model provides the best performance with accuracy and an F1 score of 90.34% with 93.79% for precision. The $BERT_{base}$-CNN-BiGRU-ATT model obtains a performance of 88.49% for accuracy, 91.07% for precision, 94.14% for recall, and 92.58% for the F1 score. The last proposed $BERT_{base}$-CNN-BiGRU model yields an accuracy of 90.10%, precision of 90.73%, recall of 96.91%, and an F1 score of 93.72%. Summaries of the prediction results of the models in the GossipCop dataset are shown in the forms of confusion matrices in Figures 8–11, showing the overall scores across all of the models.

We were also eager to further assess the effectiveness of the proposed architectures. Therefore, we conducted a case study where we tested the model's performance by ignoring user posting behavior features and comparing the model's performance. We observed that when neglecting user posting behavior information, the performance deteriorated in the two datasets. The experimental results are tabulated in Tables 4 and 5 for the PolitiFact and GossipCop datasets, respectively. The best-performing model is shown to be $BERT_{base}$-CNN-BiGRU in the two datasets. The $BERT_{base}$-CNN-BiGRU-ATT model obtained a performance of 87.77% for accuracy, 86.21% for precision, 93.75% for recall, and 89.82% for the F1 score in the PolitiFact dataset, lower than the results obtained by existing studies [36,37], while the model achieved an accuracy of 85.84%, precision of 88.49%, recall of 93.16%, and 90.98% for the F1 score in the GossipCop dataset. The $BERT_{base}$-BiGRU-CNN-ATT model yielded an accuracy of 88.49%, and precision, recall, and F1 score of 86.44%, which is on par with [36,37] in the Politifact dataset and higher than the official baselines in the GossipCop dataset. Summaries of the prediction results of the models in the PolitiFact and GossipCop datasets are shown in the forms of confusion matrices in Figures 12–14 and Figures 15–17, respectively. Figures 18 and 19 show the overall scores across all of the models in both datasets, respectively.

**Table 2.** Comparison (%) of the results obtained in the PolitiFact dataset using news content and user posting behavior features.

| Model | Accuracy (%) | Precision (%) | Recall (%) | F1 (%) |
|---|---|---|---|---|
| SAF [19] | 0.691 | 0.638 | 0.789 | 0.706 |
| BiLSTM-BERT [36] | 0.885 | NA | NA | NA |
| LNN-KG [37] | 0.880 | 0.9011 | 0.880 | 0.8892 |
| Logistic Regression (N-Gram) [38] | 0.80 | 0.79 | 0.78 | 0.78 |
| dEFEND [20] | 0.904 | 0.902 | 0.956 | 0.928 |
| BERT$_{base}$-CNN-BiGRU-ATT | 0.9281 | 0.9375 | 0.9375 | 0.9375 |
| BERT$_{base}$-BiGRU-CNN-ATT | 0.8705 | 0.8690 | 0.9125 | 0.8902 |
| BERT$_{base}$-CNN-BiGRU | 0.9137 | 0.9359 | 0.9125 | 0.9241 |

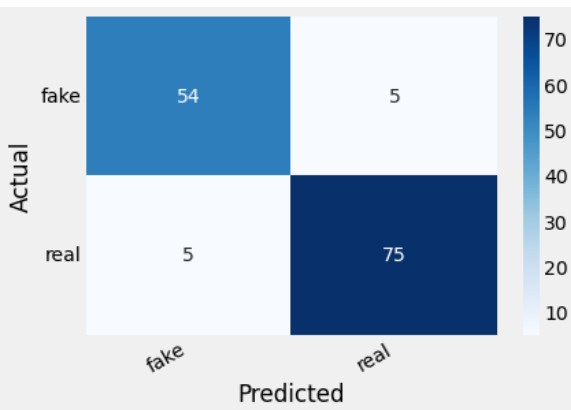

**Figure 4.** Confusion matrix of BERT$_{base}$-CNN-BiGRU-ATT in PolitiFact.

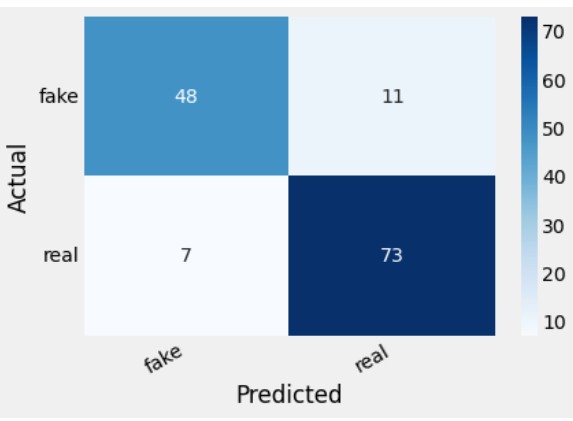

**Figure 5.** Confusion matrix of BERT$_{base}$-BiGRU-CNN-ATT in PolitiFact.

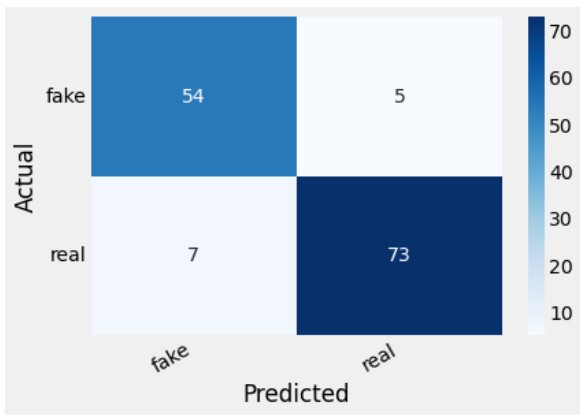

**Figure 6.** Confusion Matrix of BERT$_{base}$-CNN-BiGRU in PolitiFact.

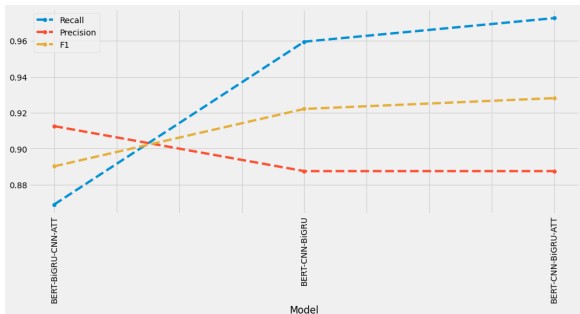

**Figure 7.** Performance comparisons (%) of all models in PolitiFact.

**Table 3.** Comparison (%) of the results obtained in the GossipCop dataset using news content and user posting behavior features.

| Model | Accuracy (%) | Precision (%) | Recall (%) | F1 (%) |
|---|---|---|---|---|
| SAF [19] | 0.796 | 0.820 | 0.753 | 0.785 |
| Logistic Regression (N-Gram) [38] | 0.82 | 0.75 | 0.79 | 0.77 |
| dEFEND [20] | 0.808 | 0.729 | 0.782 | 0.755 |
| BERT$_{base}$-CNN-BiGRU-ATT | 0.8849 | 0.9107 | 0.9414 | 0.9258 |
| BERT$_{base}$-BiGRU-CNN-ATT | 0.9034 | 0.9197 | 0.9568 | 0.9379 |
| BERT$_{base}$-CNN-BiGRU | 0.9010 | 0.9073 | 0.9691 | 0.9372 |

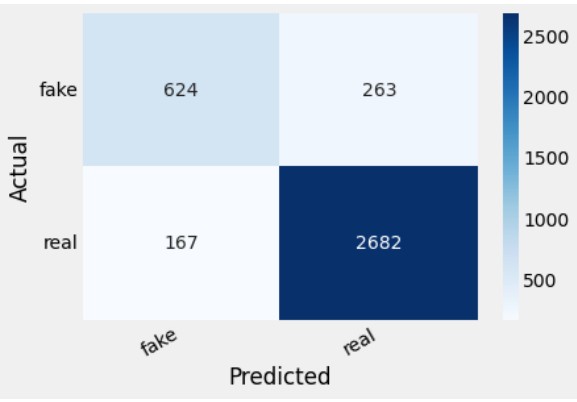

**Figure 8.** Confusion matrix of BERT$_{base}$-CNN-BiGRU-ATT in GossipCop.

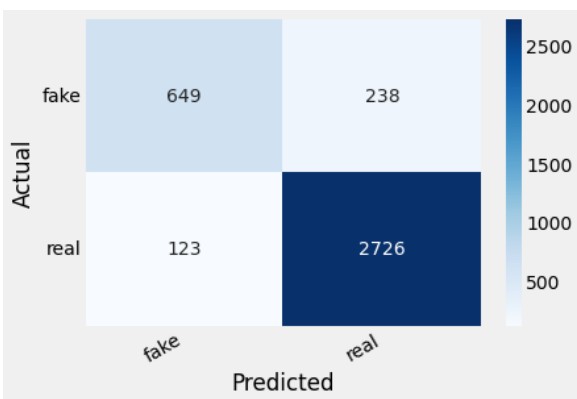

**Figure 9.** Confusion Matrix of BERT$_{base}$-BiGRU-CNN-ATT in GossipCop.

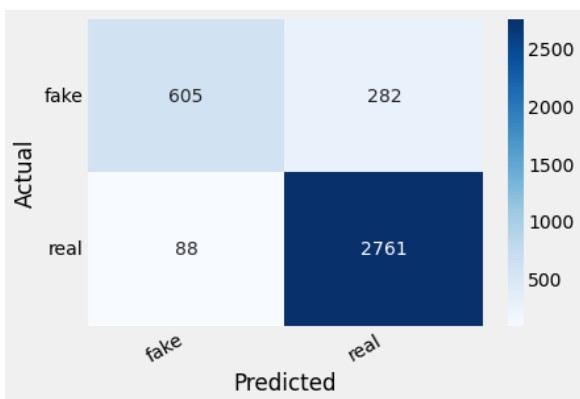

**Figure 10.** Confusion matrix of BERT$_{base}$-CNN-BiGRU in GossipCop.

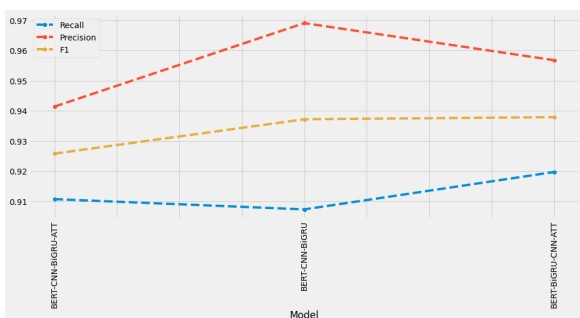

**Figure 11.** Performance comparisons (%) of all models in GossipCop.

**Table 4.** Comparison (%) of the results obtained in the PolitiFact dataset w/o user posting behavior features.

| Model | Accuracy (%) | Precision (%) | Recall (%) | F1 (%) |
|---|---|---|---|---|
| SAF [19] | 0.691 | 0.638 | 0.789 | 0.706 |
| BiLSTM-BERT [36] | 0.885 | NA | NA | NA |
| LNN-KG [37] | 0.880 | 0.9011 | 0.880 | 0.8892 |
| Logistic Regression (N-Gram) [38] | 0.80 | 0.79 | 0.78 | 0.78 |
| dEFEND [20] | 0.904 | 0.902 | 0.956 | 0.928 |
| BERT$_{base}$-CNN-BiGRU-ATT | 0.8777 | 0.8621 | 0.9375 | 0.8982 |
| BERT$_{base}$-BiGRU-CNN-ATT | 0.8849 | 0.8644 | 0.8644 | 0.8644 |
| BERT$_{base}$-CNN-BiGRU | 0.8993 | 0.9342 | 0.8875 | 0.9103 |

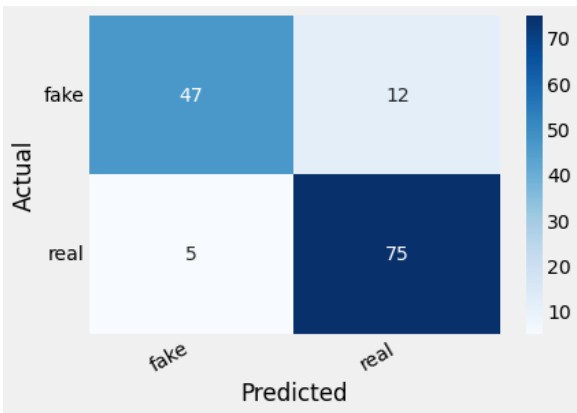

**Figure 12.** Confusion matrix of BERT$_{base}$-CNN-BiGRU-ATT in PolitiFact w/o user posting behavior features.

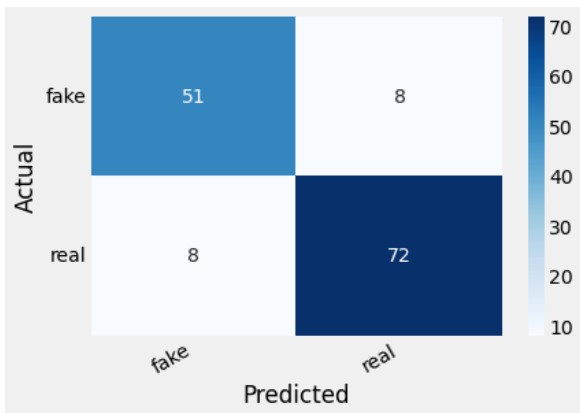

**Figure 13.** Confusion matrix of BERT$_{base}$-BiGRU-CNN-ATT in PolitiFact w/o user posting behavior features.

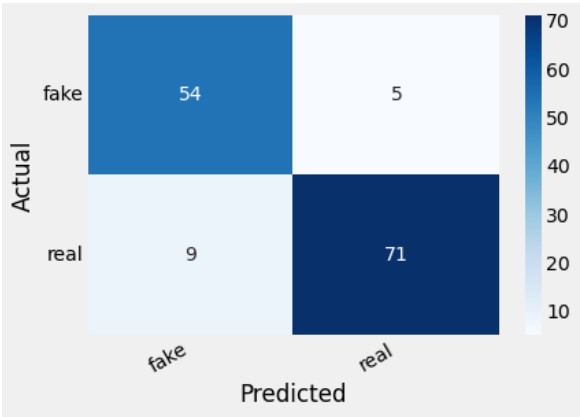

**Figure 14.** Confusion matrix of BERT$_{base}$-CNN-BiGRU in PolitiFact w/o user posting behavior features.

**Table 5.** Comparison (%) of the results obtained in the GossipCop dataset w/o user posting behavior features.

| Model | Accuracy (%) | Precision (%) | Recall (%) | F1 (%) |
|---|---|---|---|---|
| SAF [19] | 0.796 | 0.820 | 0.753 | 0.785 |
| Logistic Regression (N-Gram) [38] | 0.82 | 0.75 | 0.79 | 0.77 |
| dEFEND [20] | 0.808 | 0.729 | 0.782 | 0.755 |
| BERT$_{base}$-CNN-BiGRU-ATT | 0.8584 | 0.8849 | 0.9361 | 0.9098 |
| BERT$_{base}$-BiGRU-CNN-ATT | 0.8501 | 0.8902 | 0.9165 | 0.9031 |
| BERT$_{base}$-CNN-BiGRU | 0.8619 | 0.8619 | 0.9751 | 0.9150 |

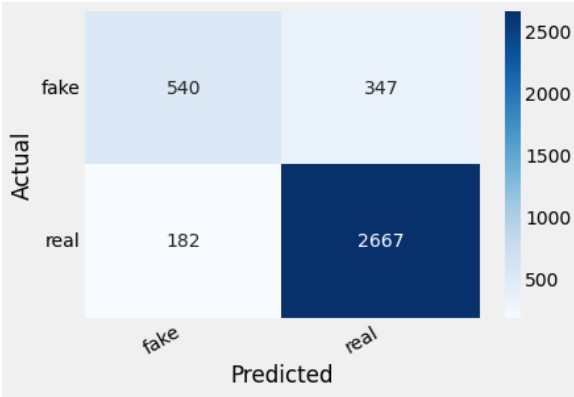

**Figure 15.** Confusion matrix of BERT$_{base}$-CNN-BiGRU-ATT in GossipCop w/o user posting behavior features.

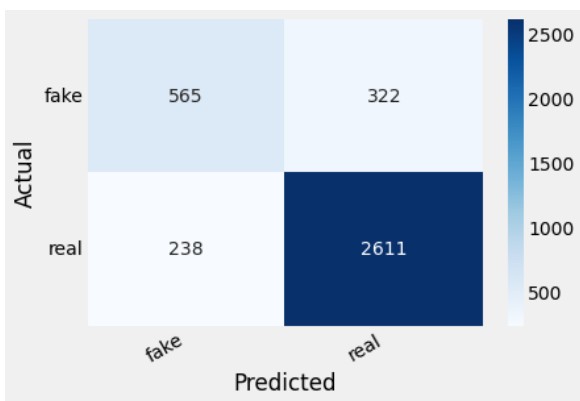

**Figure 16.** Confusion matrix of BERT$_{base}$-BiGRU-CNN-ATT in GossipCop w/o user posting behavior features.

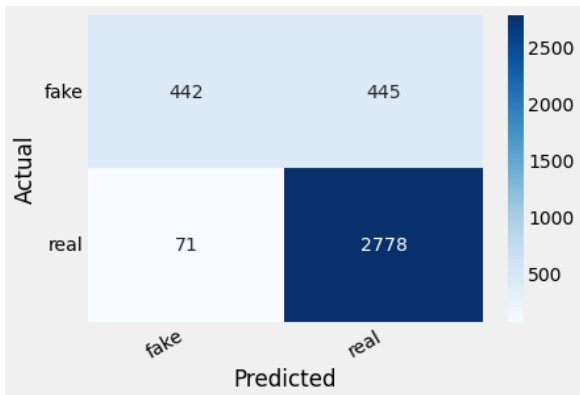

**Figure 17.** Confusion matrix of BERT$_{base}$-CNN-BiGRU in GossipCop w/o user posting behavior features.

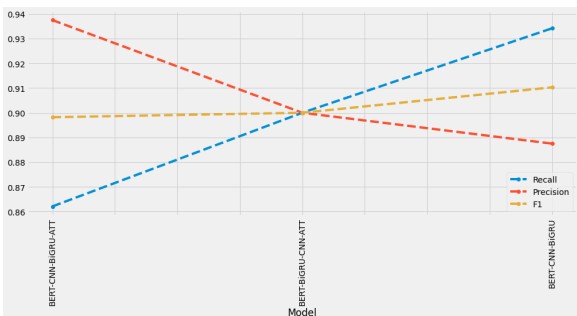

**Figure 18.** Performance comparisons (%) of models in PolitiFact w/o user posting behavior features.

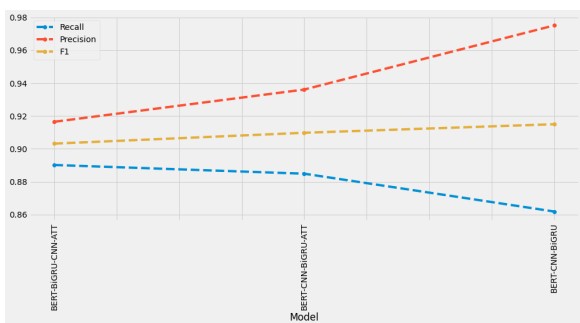

**Figure 19.** Performance comparisons (%) of models in GossipCop w/o user posting behavior features.

Based on the experimental results, we made the following observations.

1. To answer the first question raised in the introduction, based on the experiments, we observed that the interplay between the news text and the user posting behavioral features improves the detection performance, achieving state-of-the-art results. This seems to confirm our hypothesis—stated in Section 2—that user posting behavior attributes and news content contain complementary information that must be encoded and captured simultaneously in order to detect fake news.

2. To answer the second question raised in the introduction, the experimental results demonstrate that by excluding context-based features, such as user-posted behavioral cues, the performance deteriorated in the two datasets, showing the power of harnessing such clues in the modeling process.

Mining user interactions with news articles on social media can aid in the identification of false information [39]. As a result, adding pertinent information from user–news interactions would provide additional knowledge and improve detection performance. The experimental findings showed that, to a certain extent, user features could be used

to efficiently quantify detection performance. Experimental findings of two real-world datasets show how the proposed models can improve detection performance significantly. Additionally, we found that simplified versions of our models that utilized specific components only exhibited degraded performance, although still superior to baseline models, but were outperformed by the full proposed models that integrate the semantically rich information of news with user posting behavior features. The proposed model structures, on the other hand, are relatively complex. We anticipate that by using DistilBert, which has fewer parameters than BERT$_{base}$, our model will be trained efficiently in a large volume of real-world data. It has been demonstrated that combining content and context features provides discriminatory power for distinguishing fake from real content. This study emphasizes the existence of a strong and consistent correlation between news content and user behavior information when semantic contextualized relations derived from content and context features are combined for improved detection quality.

## 8. Conclusions

This study leverages context and content-based features for fake news detection. More specifically, tests were conducted to determine the impacts of considering the interplay between news content and the user posting behavior features on the performance of the detection models. The test results show that the combination of such features positively improves the detection performance. Our study exploits the power of the context-aware BERT$_{base}$ model for modeling the input text of news content. In addition, several deep learning models, such as CNN and bidirectional GRU (with attention module) were evaluated in this study using two well-known real-world fake news datasets. Based on the test results, BERT$_{base}$-CNN-BiGRU-ATT outperformed other models and achieved state-of-the-art results in the PolitiFact dataset, while BERT$_{base}$-BiGRU-CNN-ATT provided the best performance score in the GossipCop dataset.

The following will be used in the future to improve the proposed architectures:

- Other context-based features, e.g., user comments, replies, spatiotemporal features, and propagation features, can improve detection performance.
- Experiments can be conducted using other context-aware models, such as RoBERTa.
- Data augmentation techniques can be applied to balance the GossipCop dataset to reduce the potential bias and improve the detection performance.

**Author Contributions:** Conceptualization, algorithm development, experiments, and formal analysis, J.A.; writing, J.A.; writing, review, editing, analysis, interpretation of the experimental data, and supervision, Y.L. and S.L. All authors have read and agreed to the published version of the manuscript.

**Funding:** This research received no external funding.

**Institutional Review Board Statement:** Not applicable.

**Informed Consent Statement:** Not applicable.

**Data Availability Statement:** The FakeNewsNet data are available at https://github.com/KaiDMML/FakeNewsNet, accessed on 15 January 2023.

**Conflicts of Interest:** The authors declare no conflict of interest.

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
