# Peer review of "Does Context Matter? Effective Deep Learning Approaches to Curb Fake News Dissemination on Social Media"

_applsci, doi:10.3390/app13053345_

Round 1
Reviewer 1 Report
§ This paper deals with an interplay between news content and user posting behavior clues for fake news detection using state-of-the-art deep learning approaches such as Convolutional Neural Network (CNN), bidirectional Gated Recuurent Unit (BiGRU) coupled with self-attention mechanism. Some crucial issues have been highlighted in order to be considered. All the acronyms should be defined and explained first before using them such that they become evident for the readers.
§ The Introduction and related work parts give valuable information for the readers as well as researchers. Additionally, the recent papers should be added in the part of related work.
- Liu, Tong, Ke Yu, Lu Wang, Xuanyu Zhang, Hao Zhou, and Xiaofei Wu. "Clickbait detection on WeChat: A deep model integrating semantic and syntactic information." Knowledge-Based Systems 245 (2022): 108605.
- Razaque, Abdul, Bandar Alotaibi, Munif Alotaibi, Fathi Amsaad, Ansagan Manasov, Salim Hariri, Banu B. Yergaliyeva, and Aziz Alotaibi. "Blockchain-enabled deep recurrent neural network model for clickbait detection." IEEE Access 10 (2021): 3144-3163.
- Wei, Lihong, Jiankun Gong, Jing Xu, Nor Eeza Zainal Abidin, and Oberiri Destiny Apuke. "Do social media literacy skills help in combating fake news spread? Modelling the moderating role of social media literacy skills in the relationship between rational choice factors and fake news sharing behaviour." Telematics and Informatics 76 (2023): 101910.
- Razaque, Abdul, Bandar Alotaibi, Munif Alotaibi, Shujaat Hussain, Aziz Alotaibi, and Vladimir Jotsov. "Clickbait detection using deep recurrent neural network." Applied Sciences 12, no. 1 (2022): 504.
§ Representation of figures needs to be improved.
§ I'll be short. In my opinion the submitted manuscript cannot be published. Next to dozens of editing type deficiencies (indices of notations, commas, colons, dots and many more) there are serious mathematical errors.
§ The readability and presentation of the study should be further improved. The paper suffers from language problems.
§ The importance of the design carried out in this manuscript can be explained better than other important studies published in this field. I recommend the authors to review other recently developed works.
§ Please provide the link for the source code to help other scholars reproduce the experiment
§ What makes the proposed method suitable for this unique task? What new development to the proposed method have the authors added (compared to the existing approaches)? These points should be clarified.
§ Please add a Methodology section in the paper and include relevant material in this section.
§ Discussion” section should be added in a more highlighting, argumentative way. The author should analysis the reason why the tested results is achieved.
§ The authors should clearly emphasize the contribution of the study. Please note that the up-to-date of references will contribute to the up-to-date of your manuscript. Also, indicate the contribution in the “Introduction” section.
§ The complexity of the proposed model and the model parameter uncertainty are not enough mentioned.
§ It will be helpful to the readers if some discussions about insight of the main results are added.
Author Response
Dear Reviewer: Thank you for your valuable feedback.
- This paper deals with an interplay between news content and user posting behaviour clues for fake news detection using state-of-the-art deep learning approaches such as Convolutional Neural Network (CNN), bidirectional Gated Recurrent Unit (BiGRU) coupled with self-attention mechanism. Some crucial issues have been highlighted in order to be considered. All the acronyms should be defined and explained first before using them such that they become evident for the readers.
A sentence is added to explain each acronym.
In this work, we investigate the interplay between news content and user posting behaviour clues for fake news detection by using state-of-the-art deep learning approaches such as Convolutional Neural Network (CNN)--- A series of filters of different sizes and shapes combine the original sentence matrix to create further low-dimensional matrices, bidirectional Gated Recurrent Unit (BiGRU)--- A type of bidirectional recurrent neural networks with only the input and forget gates, coupled with self-attention mechanism.
- The Introduction and related work parts give valuable information for the readers as well as researchers. Additionally, the recent papers should be added in the part of related work.
Added in the related work; highlighted in red.
- Representation of figures needs to be improved.
We have redrawn and improved the figures.
- I'll be short. In my opinion the submitted manuscript cannot be published. Next to dozens of editing type deficiencies (indices of notations, commas, colons, dots and many more) there are serious mathematical errors.
The manuscript has been revised including the mathematical notations, commas, colons and dots and we couldn’t find errors.
- The readability and presentation of the study should be further improved. The paper suffers from language problems.
Improved
- The importance of the design carried out in this manuscript can be explained better than other important studies published in this field. I recommend the authors to review other recently developed works.
We reviewed other recently developed works in the related work section (i.e., highlighted in red) and we clarify the limitation of the existing work that we addressed.
- Please provide the link for the source code to help other scholars reproduce the experiment
The link is provided in the Methodology section.
- What makes the proposed method suitable for this unique task? What new development to the proposed method have the authors added (compared to the existing approaches)? These points should be clarified.
We clarified this by mentioning the components of the proposed architectures where we used in the third model the interplay between the deep learning modules (CNN and BiGRU) and the pooled output of BERT model. We also investigate the effectiveness of different models. For detecting fake news, we used context-aware representations instead of context-independent representations heavily used by previous/existing related works.
- Please add a Methodology section in the paper and include relevant material in this section.
The Methodology section is added.
- Discussion” section should be added in a more highlighting, argumentative way. The author should analysis the reason why the tested results is achieved.
The Discussion section is added.
- The authors should clearly emphasize the contribution of the study. Please note that the up-to-date of references will contribute to the up-to-date of your manuscript. Also, indicate the contribution in the “Introduction” section.
Up-to-date references are added in the related work and the key contributions are added and highlighted in the introduction.
- The complexity of the proposed model and the model parameter uncertainty are not enough mentioned.
A paragraph that explains the complexity of the models is added in the Discussion section and highlighted in red.
- It will be helpful to the readers if some discussions about insight of the main results are added.
Added in the results section.

Reviewer 2 Report
There are some mistakes in this paper that must be incorporated in the resubmission document.

Author Response
Dear Reviewer,
Thank you for your valuable feedback.
There are some mistakes in this paper that must be incorporated in the resubmission document.
The manuscript has been revised and improved.

Round 2
Reviewer 1 Report
The authors tried to respond the comments and I recommend for acceptance
Author Response
We have proofread the paper and fixed some minor editing errors

Reviewer 2 Report
minor changes required

Author Response
Dear Reviewer,
Thank you very much for your feedback.
- In the abstract section the authors used BERT whereas in section 3, BERTBase is used. Explain the difference.
We have added the following paragraph to explain the difference. And we updated the keyword BERT throughout the manuscript.
Even though BERT contains millions of parameters (i.e., BERT$_{base}$ contains 110 million parameters and BERT$_{large}$ has 340 million parameters)\cite{devlin2018bert}, it is relatively inexpensive to apply BERT to downstream tasks if the parameters are fine-tuned using a pre-trained model. In this work, we use BERT$_{base}$.
- Equation shown in section 4 must be numbering like the other equations.
Modified
- Figures 4,5,6 are similar and the text in these figures is also similar. Then why shown these three graphs. Explain it.
Added in the Results section, highlighted in red. The confusion matrix is included for each method.
On PolitiFact, all models showed similar confusion matrix results, with BERTbase-CNN-BiGRU-ATT yielding the best results.
4. The title of figure 7 is overlapping. Rewrite it with clear text.
Modified
5. Section 7 and 8 must be combine together.
Combined
We have proof read the paper and fixed some minor editing errors

Round 3
Reviewer 2 Report
Minor changes in the above-mentioned paper
